# Equipment of Vertically-Ordered Mesoporous Silica Film on Electrochemically Pretreated Three-Dimensional Graphene Electrodes for Sensitive Detection of Methidazine in Urine

**DOI:** 10.3390/nano13020239

**Published:** 2023-01-05

**Authors:** Xiaochun Deng, Xueting Lin, Huaxu Zhou, Jiyang Liu, Hongliang Tang

**Affiliations:** 1Department of Chemistry, Key Laboratory of Surface & Interface Science of Polymer Materials of Zhejiang Province, Zhejiang Sci-Tech University, Hangzhou 310018, China; 2The First Clinical Faculty of Guangxi University of Chinese Medicine, Nanning 530200, China; 3Affiliated Fangchenggang Hospital, Guangxi University of Chinese Medicine, Fangchenggang 538001, China

**Keywords:** electrochemical sensor, three-dimensional electrode, macroporous graphene foam, vertically-ordered mesoporous silica film, methidazine

## Abstract

Direct, rapid, and sensitive detection of drugs in complex biological samples is essential for drug abuse control and health risk assessment. In this work, an electrochemical sensor was fabricated based on equipment of vertically-ordered mesoporous silica film (VMSF) on an electrochemically pre-treated three-dimensional graphene electrode (p-3DG), which can achieve direct and sensitive determination of methylthiopyridazine (TR) in urine. Three-dimensional graphene (3DG) with a continuous and interpenetrating graphene network was used as the supporting electrode and simple electrochemical polarization was employed to pre-treat 3DG to improve surface hydrophilicity and electrocatalytic performance. VMSF was easily grown using an electrochemical assisted self-assembly method within 10 s and was stably bound to the p-3DG surface. The nanochannel array on the as-prepared VMSF/p-3DG sensor enriched positively charged TR, leading to significantly improved electrochemical signal. Combined with the high electric activity of p-3DG and the enrichment of nanochannels, VMSF/p-3DG realized sensitive determination of TR ranging from 50 nM to 10 μM with a low detection limit (DL, 30 nM). Owing to the anti-fouling and anti-interference performance of VMSF, the common electroactive molecules including ascorbic acid (AA) and uric acid (UA) did not interfere with the detection. In addition, the detection of TR in buffer and urine exhibited similar sensitivity. Accurate detection of TR in urine was realized.

## 1. Introduction

Drug abuse can lead to drug addiction and other behavioral disorders, causing serious public health and social problems. For instance, thioridazine (TR) is an active drug commonly used to treat schizophrenia and mania, as well as depression, menopausal syndrome, epileptic psychosis, senile psychosis, etc. [1]. Taking TR may cause some side effects in the organism, such as somnolence, headache, blurred vision, and dizziness. In addition, excessive use of TR may increase serum prolactin level [2], leading to ovulation dysfunction and infertility. Unexplained deaths due to overdose of TR are also reported by Reilly et al. [3]. Therefore, sensitive detection of TR in biological samples plays a vital role in preventing the abuse of TR of biological samples. Until now, the technologies used to detect TR have included fluorescence sensors [4], flow injection coupled with fluorescence [5], ultraviolet–visible (UV-Vis) spectrophotometry [6], supercritical fluid chromatography (SFC) [7], colorimetric detection [8], gas chromatography (GC) [9], and high-performance liquid chromatography (HPLC) [10]. However, these approaches usually need time-consuming and complex sample pretreatment, expensive instruments, and professional operators. In comparison, electrochemical sensors are highly attractive because of their advantages, such as fast detection, convenient operation, cheap instrument, easy integration, and portability [11,12,13,14]. The combination of functional nanomaterials can further improve detection performance. For example, Mashhadizadeh et al. [15] used a zinc sulfide (ZnS) nanoparticles modified carbon paste electrode (ZnSNP/MCPE) to detect TR. Sakthivel et al. [16] prepared a ruthenium-doped bismuth sulfide (Ru-Bi_2_S_3_) modified glassy carbon electrode (GCE) that exhibited sensitive response towards TR. Feng et al. [17] modified the GCE with nitrogen-doped carbon nanotubes/gold nanoparticle (N-CNTs/Au) composite for electrochemical detection of TR. However, the employed nanomaterials commonly suffer from a complex synthesis process and the electrode is easy to be contaminated, which still limits the application for analysis of complex biological samples (e.g., serum, urine, etc.). Therefore, development of electrochemical sensors with a simple preparation process, high sensitivity, and good anti-interference/anti-fouling ability is highly desirable for direct detection of TR in biological samples.

Porous materials have recently attracted great attention in the fields of drug delivery, adsorption, catalysis, and sensing owing to merits of adjustable structure, porous structure, and high surface area. Integrating porous nanomaterials with size or charge screening capability on the electrode surface can efficiently improve the anti-fouling and anti-interference performance of electrochemical sensors [18,19]. Recent research has found that vertically-ordered mesoporous silica film (VMSF) modified electrode has great potential in direct electroanalysis of complex samples [20,21,22]. VMSF has a highly ordered nanochannel array with uniform pore size (commonly 2–3 nm) and high porosity (as high as 7.5 × 10^12^ pores/cm^2^) that is perpendicular to the underlying electrode [23,24]. At present, VMSF can be rapidly grown (within 10 s) on the conductive substrate via an electrochemically assisted self-assembly method (EASA) [25]. Large-area VMSF (dozens or hundreds of cm^2^) can also be prepared using the Stöber solution growth method [26]. In addition, the ultrasmall nanochannels possess a significant size-exclusion effect. For instance, biological macromolecules or large particles such as proteins, DNA, or cells cannot enter the channels, so as not to contaminate the supporting electrode [27]. In addition, VMSF is rich in silanol groups (p*K*_a_~2). Thus, the surface of VMSF nanochannels is negatively charged in conventional pH solutions, which can generate electrostatic attraction towards positively charged small molecules and electrostatic repulsion to negatively charged small molecules [28,29,30]. This phenomenon will eliminate the influence of negative charged interferences while enriching positive analytes. Therefore, VMSF has the advantages of simple preparation, selective enrichment of specific analytes, and excellent anti-interference/anti-fouling abilities [31]. However, VMSF is difficult to combine stably on commonly used electrochemical electrodes, such as glassy carbon electrodes or gold electrodes. Therefore, it is important to equip electrodes with stable VMSF for direct electroanalysis of TR in biological samples.

Graphene-based nanomaterials contain 0D graphene quantum dots [32,33], 2D graphene nanosheets [34], 3D porous graphene [35], and so on. Owing to unique optical, electrical, or catalytic properties, the graphene family commonly displays great potential for the fabrication of a novel sensing platform with good performance [36,37,38]. Amongst these, graphene is a two-dimensional (2D) nanocarbon material with honeycomb structure connected by sp^2^ carbon atoms that has single-layered thickness, excellent electron transfer rate, and high theoretical specific surface area. Because of the strong π-π interaction between graphene sheets, however, graphene sheets are easy to stack, resulting in the reduction of surface area or active sites and limited application in electroanalysis [39]. Three-dimensional (3D) graphene is graphene material with a 3D porous structure that can effectively overcome the stacking of 2D graphene sheets [40,41,42]. Commonly, three-dimensional graphene foam (3DG) can be synthesized via chemical vapor deposition (CVD) when nickel foam is applied as the macroporous template. This 3DG is the most common three-dimensional graphene material and has been commercially produced [43]. Specifically, 3DG is a continuous and interpenetrating graphene network with high electron transport rate, large specific surface area, and good mass transfer performance. However, the high hydrophobicity of 3DG hinders its application in biological analysis. Using simple methods to increase the hydrophilicity of 3DG and realize the integration of VMSF on its surface is critical to expand the application of 3DG in direct electroanalysis of complex biological samples.

In this work, an electrochemical sensor was easily fabricated by integrating VMSF on the electrochemically pre-activated three-dimensional graphene electrode (p-3DG), which can be applied for direct and sensitive electrochemical detection of TR in urine. The 3DG was pretreated using a simple electrochemical polarization method to increase its hydrophilicity through introduction of oxygen-containing groups. Then, VMSF was rapidly grown on the p-3DG via the EASA method, realizing the stable binding of VMSF without an adhesive layer. Owing to the electrostatic enrichment of TR by VMSF nanochannels, the VMSF/p-3DG sensor exhibits high sensitivity for the detection of TR. Combined with the anti-fouling and anti-interference abilities of VMSF, direct and highly sensitive detection of TR in urine is achieved.

## 2. Materials and Methods

### 2.1. Chemicals and Materials

Thioridazine (TR), 1-butyl-3-methylimidazolium hexafluorophosphate (BMIMPF_6_), tetraethoxysilane (TEOS, 98%), potassium ferricyanide (K_3_Fe(CN)_6_), disodium hydrogen phosphate heptahydrate (Na_2_HPO_4_·7H_2_O), sodium dihydrogen phosphate (NaH_2_PO_4_), and potassium hydrogen phthalate (KHP) were obtained from Shanghai Aladdin Reagent Co., Ltd. (Shanghai, China). Hexadecyl trimethyl ammonium bromide (CTAB) and acetonitrile (99.9%) were bought from Shanghai Maclin Biochemical Technology Co., Ltd. (Shanghai, China). Hydrochloric acid (HCl) and ethanol were gained from Hangzhou Shuanglin Chemical Reagent Co., Ltd. (Hangzhou, China). The reagents were all analytically pure and do not require further purification. The aqueous solutions were prepared with ultra-pure water (18.2 MΩ·cm).

### 2.2. Measurements and Instrumentations

The morphology was characterized via scanning electron microscope (SEM) and transmission electron microscope (TEM). SEM measurement was performed with a SU8010 microscope (SEM, Hitachi Co., Tokyo, Japan) at an accelerated voltage of 10 kV. TEM was performed on a JEM-2100 microscope (JEOL Ltd., Tokyo, Japan) with an acceleration voltage of 200 kV. X-ray photoelectron spectroscopy (XPS) spectra were obtained on a PHI5300 (USA, Perkin Elmer, Waltham, MA, USA) with Mg Kα being the excitation source. All electrochemical experiments were carried out on an Autolab Electrochemical Station (PGSTAT302N, Metrohm, Herisau, Switzerland). In cyclic voltammetry (CV) and differential pulse voltammetry (DPV) measurements, a simple three-electrode system was adopted. Briefly, 3DG, p-3DG, or VMSF/p-3DG were used as the working electrodes. Platinum wire or platinum sheet was employed as the counter-electrode. For the nonaqueous experiment, a Ag/Ag^+^ (10 mM Ag^+^/acetonitrile solution) electrode was used as the reference electrode. For aqueous electrolyte, Ag/AgCl (saturated KCl solution) was employed as the reference electrode. DPV parameters included a step potential of 5 mV, pulse amplitude of 25 mV, pulse time of 0.05 s, and time interval of 0.2 s.

### 2.3. Preparation of 3DG Electrode and p-3DG Electrode

According to previous reports [44], 3DG was produced via chemical vapor deposition (CVD) under atmospheric pressure using nickel foam as a template and ethanol as a carbon source. After the synthesized material was immersed in HCl solution (3 M) at 80 °C for 24 h, the free-standing 3DG without the nickel template was obtained. To make the electrode, glass sheet was applied as the supporting substance and copper wire was used as the connecting wire. Briefly, 3DG foam (0.5 cm × 0.5 cm) was fixed on the glass sheet with silica gel. The conductive silver glue served as bridge links to connect the 3DG with the copper wire. After sealing the conductive silver glue and the copper wire with insulating silica gel, the 3DG electrode was obtained.

The 3DG electrode was then electrochemically polarized by anodic oxidation and cat hodic reduction to produce the pre-treated 3DG electrode (p-3DG) [45]. Specifically, the acetonitrile solution containing BMIMPF_6_ ionic liquid (10%, *V*/*V*) was used as the anodized oxidation electrolyte. To perform anodic oxidation, the 3DG electrode was used as the working electrode and a constant voltage of +5 V was applied for 100 s. Then, phosphate buffer solution (PBS, 0.1 M, pH 6) was used as the electrolyte and a constant voltage of −1 V was applied for 300 s for cathode reduction [46,47]. After being thoroughly rinsed, p-3DG was obtained.

### 2.4. Preparation of VMSF/p-3DG Electrode

VMSF was grown on p-3DG electrode (VMSF/p-3DG) via the electrochemically assisted self-assembly method [25,44,48]. Briefly, the mixture containing CTAB (1.585 g), TEOS (3050 μL), and ethanol (20 mL) was prepared in NaNO_3_ (20 mL, 0.1 M, pH 2.6). The obtained solution was pre-hydrolyzed under agitation for 2.5 h to produce the precursor solution for VMSF growth. Three steps were involved for the preparation of VMSF/p-3DG. Firstly, the p-3DG electrode was soaked in the prepared solution and reacted for 10 s under a constant current density of −0.74 mA/cm^2^. Then, the obtained electrode was removed promptly and completely cleaned with ultrapure water. Secondly, the resulting electrode was aged overnight at 80 °C to obtain the SM-blocked electrode (SM@VMSF/p-3DG). Thirdly, the SM@VMSF/p-3DG electrode was immersed in ethanol/hydrochloric acid (HCl) solution with stirring for 5 min to eliminate the SM. Finally, a VMSF/p-3DG electrode with open nanochannel array was obtained.

### 2.5. Electrochemical Detection of TR

The electrochemical detection of TR was performed in PBS (0.1 M, pH = 7.4) electrolyte. The DPV curves were recorded on VMSF/p-3DG after adding different concentrations of TR. The scanned potential ranged from 0.3 V to 1.0 V. For real sample analysis, TR in urine was measured using standard addition method. Before determination, urine (health man) with artificially added TR was diluted by a factor of 50 times using PBS electrolyte.

## 3. Results and Discussion

### 3.1. Strategy for Enquipment VMSF on Electrochemical Pre-Treated 3DG

Consisting of porous and continuous graphene monolith, 3DG provides a special and steady macroporous scaffold with high conductivity, large surface area, and good mass transfer. Improving the hydrophilicity and electrocatalytic properties of 3DG is crucial to explore its application in the field of electroanalysis. It has been reported that treatment of a 3DG electrode with air or oxygen plasma can enhance its hydrophilicity. However, this pre-treatment requires special instruments and complex operation, and the electrocatalytic activity of the treated 3DG has not been significantly improved. As illustrated in Figure 1, simple electrochemical polarization was employed to produce the electrochemically pre-treated 3DG (p-3DG) in this work. Electrochemical polarization has been evidenced to be a convenient and practical pretreatment strategy that can significantly promote the electrocatalytic activity and potential resolution of carbon-based electrodes, thus improving detection sensitivity and selectivity [46,47]. Commonly, electrochemical polarization usually consists of anodic polarization (electrochemical oxidation) at high positive voltages (usually >1.5V) and cathodic polarization (electrochemical reduction) at negative voltages. In the process of anodic polarization, the carbon structure is significantly etched and oxygen-containing groups (e.g., hydroxyl, carboxyl, carbonyl groups) are introduced [45]. For a graphene electrode, the etching of its base surface during anodic polarization will further boost the formation of new edge surface, which can act as the electrocatalytic sites. In the subsequent electrochemical reduction process of cathodic polarization, some functional groups, such as oxygen-containing groups, will be reduced. For instance, the carbonyl group will be reduced to the hydroxyl group. In addition, the conductivity of the electrode will be restored at the same time. Therefore, electrochemical polarization can improve the hydrophilicity of 3DG and introduce electrocatalytic sites to improve the detection sensitivity and potential resolution of the electrode. Compared with other electrode modification process, electrochemical polarization has the advantages of a simple instrument and convenient operation.

As shown in Figure 1, VMSF was then grown on p-3DG via the EASA method, which is widely used because of its simple operation and short preparation time (usually within 10 s). Specifically, the negative potential is applied on p-3DG to electrolyze water, causing local pH to rise near the electrode surface. This phenomenon not only induces the self-assembly of cationic surfactant (CTAB) into surfactant micelles (SM), but also catalyzes the hydrolysis and condensation of precursor molecules (TEOS) at the interface between SM and p-3DG. Finally, vertically-aligned silica nanochannels filled with SM templates were grown on the p-3DG surface (SM@VMSF/p-3DG). Owing to the presence of functional oxygen-containing groups on the surface of p-3DG, VMSF can be stably bound to p-3DG because these functional groups (e.g., -OH) can react with the silanol group to form covalent bonds (O-Si-O). VMSF/p-3DG with open nanochannels was obtained via solvent extraction in HCl/ethanol solution to remove CTAB micelles. Compared with the surface treatment of the 3DG electrode with air plasma to enhance hydrophilicity, this method does not require special instruments or complex operations [48]. Compared with the strategy of using 3-aminopropyltriethoxysilane as the binding layer to achieve stable equipment of VMSF on the 3DG surface [49], this method does not require a non-conductive bonding layer. Therefore, the strategy of pre-activating 3DG using simple electrochemical polarization has the advantages of simple method, easy operation, improved hydrophilicity and electrocatalytic activity of 3DG, and direct growth of VMSF on p-3DG without an adhesive layer.

### 3.2. Characterization of 3DG and p-3DG

The electrochemical properties of 3DG and p-3DG were studied via cyclic voltammetry using Fe(CN)_6_^3−^ as an electroactive probe. As shown in Figure 2a, p-3DG exhibits larger charging current, and the redox peak of Fe(CN)_6_^3−^ on p-3DG is obviously larger than that on 3DG, indicating increased electrochemical performance. The hydrophilicity of 3DG and p-3DG are also characterized by measurement of the contact angle. As shown in Figure 2b, the contact angle of 3DG is 133.9°, indicating high hydrophobicity. On the contrary, p-3DG exhibits a significantly decreased contact angle of 50.9° suggesting remarkably increased hydrophilicity. The changes of chemical groups on the electrode surface before and after electrochemical polarization were characterized via X-ray photoelectron spectroscopy (XPS). Figure 2c shows the high-resolution C1s spectrum of the bare 3DG electrode. As shown, the 3DG electrode has a high content of C-C/C=C (284.6 eV), indicating the sp^2^ carbon structure of graphene materials. A small amount of oxygen containing functional groups might come from the oxygen adsorbed on the graphene surface. Compared with the 3DG electrode, the C-C/C=C content of p-3DG decreases while the content of C-O (286.1 eV), C=O (287.7 eV) and O-C=O (288.9 eV) increases, indicating rich oxygen-containing functional groups after electrochemical polarization (Figure 2d). This surface change leads to a significant improvement of hydrophilicity. The morphology of 3DG and VMSF/p-3DG were characterized using SEM. As shown in Figure 2e, 3DG presents a monolith with a macroporous graphene network and smooth surface. The high-resolution scanning electron microscope (HRSEM) image of 3DG (inset of Figure 2e) displays the wrinkled structure of graphene. In the case of p-3DG, the surface becomes rough with a few cracks (Figure 2f). These cracks might result from the anodic etching of graphene at high potential in electrochemical polarization.

### 3.3. Characterization of VMSF Modified p-3DG

The film integrity and charge-selective permeability of VMSF on a VMSF/p-3DG electrode were characterized via CV using standard anionic or cationic electrochemical probes. Figure 3 displays CV curves obtained on p-3DG, SM@VMSF/p-3DG, and VMSF/p-3DG in the presence of Fe(CN)_6_^3−^ (Figure 3a) or Ru(NH_3_)_6_^3+^ (Figure 3b). In comparison with p-3DG, the peak currents measured on SM@VMSF/p-3DG are very low, which might be ascribed to the blocked mass transfer of the redox probe resulting from the filling nanochannels by SM. This demonstrates that the VMSF film on p-3DG is intact with no defects. When SM is removed, VMSF/p-3DG exhibits significant redox peaks of both probes. In comparison with that on p-3DG, however, the peak current on VMSF/p-3DG electrode decreases in Fe(CN)_6_^3−^ solution but increases in Ru(NH_3_)_6_^3+^ probe solution, indicating charge-selective permeability. As known, VMSF has a silica structure with rich silanol groups (p*K_a_* ~ 2). Owing to the deprotonation of silanol groups at measured conditions, VMSF exhibits a negatively charged surface and displays electrostatic repulsion towards a negatively charged probe as well as electrostatic enrichment towards a positively charged probe.

Transmission electron microscopy (TEM) was employed to study the morphology of VMSF. Figure 4a shows the top-view TEM of VMSF at different magnifications. It can be seen that the film is flat and crack-free in the submicron range, and a regular arrangement of high-density pores is observed. The high-resolution TEM (HRTEM) image (inset in Figure 4b) can clearly observe the mesoporous structure of an average pore size of 2–3 nm, where the calculated porosity is about 44%. Figure 4b shows the TEM image of the cross-sectional surface of VMSF. The channel length is about 95 nm.

### 3.4. Enhanced Electrochemical Performance of TR on VMSF/p-3DG

To verify the feasibility of using the VMSF/p-3DG sensor to measure TR, we compared the electrochemical responses of different electrodes (3DG, p-3DG, and VMSF/p-3DG) towards TR. Figure 5a shows the CV curves of TR on the three electrodes. As seen, the electrochemical signal of TR on the p-3DG electrode is greatly increased compared to that on the 3DG electrode because the electrochemical polarization improves the hydrophilicity of the electrode. In addition, the oxidation peak potential shows a slight negative shift, indicating the electrocatalytic performance of p-3DG. When VMSF is equipped on the electrode, the electrochemical signal of TR further increased. Figure 5b exhibits the DPV curves of TR on the three electrodes. The oxidation peak current of TR on VMSF/p-3DG is the highest. As the p*K*_a_ of TR is ~9.5, it is positively charged in the electrolyte. Thus, TR can be enriched by the negatively charged nanochannels of VMSF, leading to high detection sensitivity. The reaction mechanism of TR on the electrode is shown in Figure 5c, which is an oxidation reaction involving two electrons.

As VMSF channels have an electrostatic enrichment effect on TR, enrichment time is an important parameter affecting detection sensitivity. Thus, the enrichment time for the detection of TR using a VMSF/p-3DG sensor is optimized. According to Figure 5d, with the increase in enrichment time, the DPV oxidation peak current of TR increases gradually. When the enrichment time reaches 10 min, the DPV signal tends to reach a plateau. Thus, 10 min was chosen to enrich TR before the detection.

### 3.5. Electrochemical Determination of TR in Buffer Using VMSF/p-3DG Sensor

Under the optimal conditions, electrochemical determination of TR using VMSF/p-3DG sensor was investigated. Figure 6a is the DPV curves gained in the presence of TR with different concentrations in the PBS electrolyte. As shown, with increasing concentration of TR, the oxidation peak current gradually increases. The oxidation peak current (*I*) has a good correlation with the concentration of TR (*C*) ranged from 50 nM to 10 μM (*I* = 1.12*C* + 0.426, *R*^2^ = 0.993, Figure 6b). We calculated a detection limit (DL) of 30 nM using a three signal-to-noise (S/N = 3). The low detection limit is attributed to high electrochemical area and electrocatalytic activity of p-3DG and the electrostatic enrichment effect of the nanochannels on TR. Table 1 summarizes the linear range and DL for the determination of TR via different electrochemical methods. It can be seen that the DL using the VMSF/p-3DG electrode is lower than that obtained by most sensors. This phenomenon is attributed to the excellent electrocatalytic activity and enhanced conductivity of p-3DG and the electrostatic enrichment effect of VMSF.

### 3.6. Anti-Interference and Anti-Fouling Abilities of VMSF/p-3DG Sensor

The anti-interference capacity of VMSF/p-3DG sensor was further examined. One of the common disturbance species in biological fluids, for instance metal ions (Na^+^, K^+^, Ca^2+^, Mg^2+^ and Zn^2+^) and biomolecules (ascorbic acid-AA, uric acid-UA, urea and glucose-Glu), was added into the TR solution. As depicted in Figure 6c, no significant change was observed even when the concentration of each substance was 50 times higher than TR. In addition, AA or UA, the usual molecules with electrochemical activity, also had no interference. This phenomenon results from the potential resolving power of p-3DG and the electrostatic repulsion of VMSF nanochannels towards these negatively charged molecules. Thus, the developed VMSF/p-3DG sensor displays high selectivity.

To evaluate the detection stability of the VMSF/p-3DG electrode in complex medium, TR was added to urine that was diluted by a factor of 10 times. Then, the sensor was immersed in the solution and detected at different times. For comparison, the signal obtained on p-3DG electrode was also compared (Figure 6d). As known, urine has a complex matrix with uric acid, urea, inorganic salts, possible proteins, and other substances, which easily contaminate the electrochemical electrode. After 70 min, the seventh detection using the p-3DG electrode retained only 56% of the initial signal according to Figure 6d, indicating fouling of the electrode and low stability in the complex matrix. On the contrary, 85% of the initial signal was retained for the seventh detection using the VMSF/p-3DG sensor, indicating high stability of the sensor owing to the anti-fouling ability of VMSF nanochannels. In addition, the electrochemical response of the developed sensor towards TR (1 µM) was 97.3% (*n* = 3) after a 7-day storage in air at room temperature, indicating high stability in storage.

### 3.7. Electrochemical Determination of TR in Urine

The metabolism of TR in the body mostly occurs in the liver, and most of TR and its metabolites are secreted into bile and then excreted through feces. Among the rest, about 10% of TR appears in urine. The use of a VMSF/p-3DG sensor to detect TR in urine was also investigated. Figure 7a shows the DPV responses gained on the VMSF/p-3DG sensor when different concentrations of TR in urine (diluted 50 times) were detected. It is evident that the oxidation peak increases with increasing TR concentration. A linear relationship was revealed between the oxidation peak current and TR concentration in the range of 50 nM to 10 μM (*I* = 1.10*C* + 0.300, *R*^2^ = 0.998, Figure 7b). It is noteworthy that the sensitivity of detection in urine (1.10 μA/μM) is almost the same as that in buffer (1.12 μA/μM). This phenomenon shows that a VMSF/p-3DG electrode has high anti-fouling ability in complex matrix, suggesting enormous potential in direct electrochemical detection of complex samples.

### 3.8. Reuse of VMSF/p-3DG Sensor

To regenerate the constructed VMSF/p-3DG sensor, the electrode was stirred in 0.1M hydrochloric acid/ethanol solution for 5 min to elute the enriched TR in the VMSF nanochannels. As shown in Figure 8, the reproduced sensor displays almost no electrochemical signals in the electrolyte. At the same time, the peak current for the detection of TR is similar with that of the initial signal obtained in the first detection. Thus, the sensor exhibits good regeneration performance and can be reused.

## 4. Conclusions

In summary, a 3D electrochemical sensor was developed for rapid and sensitive detection of TR in biological samples by integrating VMSF on 3DG electrodes with no need for an additive adhesive layer. The electrochemically pre-activated 3DG (p-3DG) electrode exhibits high electroactive area and excellent electrocatalytic performance. The rapid growth of VMSF on p-3DG is easily achieved using an electrochemically assisted self-assembly method within 10 s. The nanochannels of VMSF could enrich TR through electrostatic attaction, leading to remarkable enrichment and signal amplification. Owing to the ultrasmall nanochannels, VMSF endows the VMSF/p-3DG sensor with excellent antifouling ability. Sensitive detection of TR in urine was achieved. The developed VMSF/p-3DG sensor exhibits merits of simple electrode structure, easy fabrication, and excellent performance. The sensors constructed here pose a way forward for the fabrication of 3D electrochemical sensors and a VMSF-based sensing platform.

## Figures and Tables

**Figure 1 nanomaterials-13-00239-f001:**
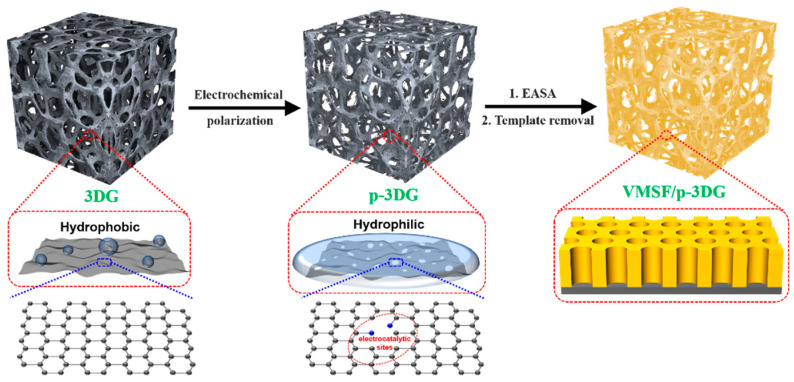
Schematic illustration for the fabrication of VMSF/p-3DG sensor.

**Figure 2 nanomaterials-13-00239-f002:**
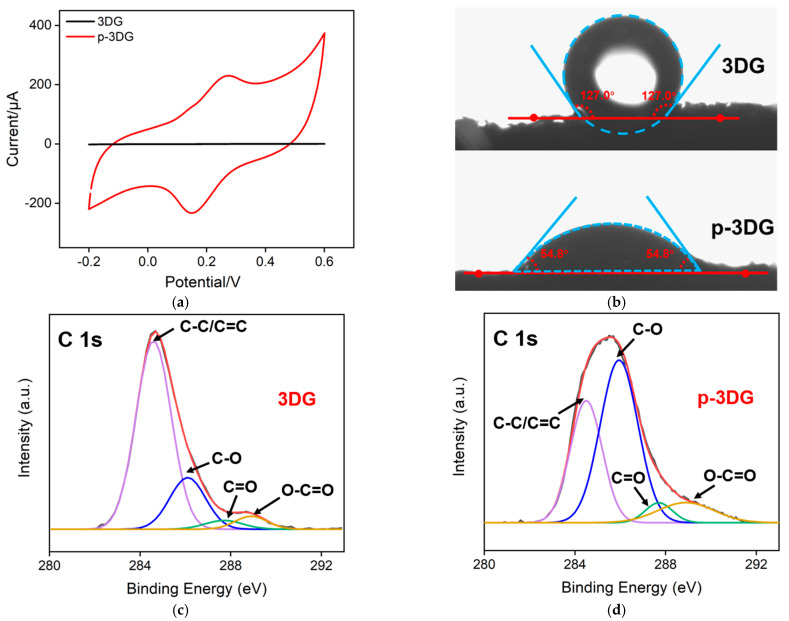
(**a**) CV responses obtained at 3DG and p-3DG electrodes in 0.5 mM Fe(CN)_6_^3−^ (in 0.1 M KCl medium). (**b**) Contact angle photos of 3DG and p-3DG. (**c**,**d**) High-resolution C 1 s XPS spectra of 3DG (**c**) and p-3DG (**d**). (**e**,**f**) SEM images of (**e**) 3DG and (**f**) VMSF/p-3DG.

**Figure 3 nanomaterials-13-00239-f003:**
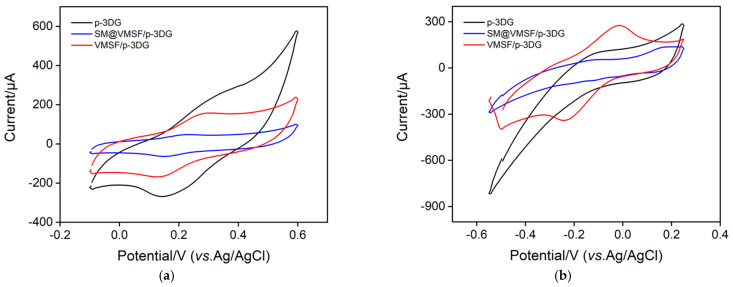
CV curves obtained on different electrodes in 0.05 M KHP solution (pH = 4) containing 0.5 mM Fe(CN)_6_^3−^ (**a**) or 0.5 mM Ru(NH_3_)_6_^3+^ (**b**).

**Figure 4 nanomaterials-13-00239-f004:**
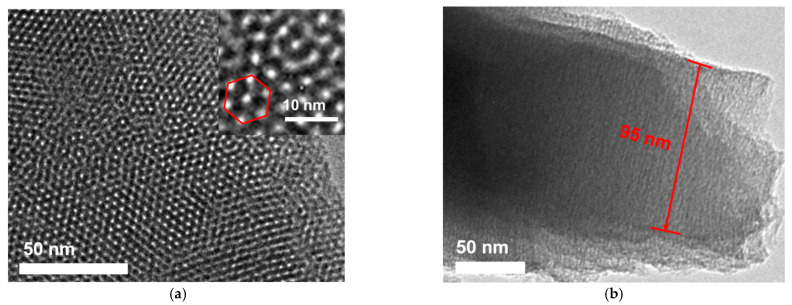
Top-view (**a**) or cross-sectional (**b**) TEM photos of VMSF. Inset in (**a**) is the high-resolution TEM (HRTEM) image.

**Figure 5 nanomaterials-13-00239-f005:**
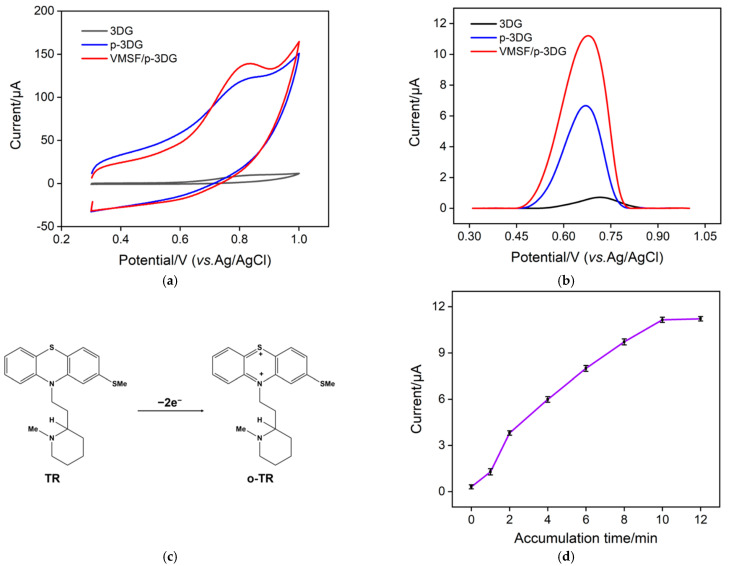
CV (**a**) and the corresponding DPV (**b**) responses gained on 3DG, p-3DG and VMSF/p-3DG electrodes in 0.01 M PBS (pH 7.4) containing TR (10 μM). (**c**) Illustration of electrochemical reaction mechanism of TR. (**d**) The oxidation peak currents measured on VMSF/p-3DG towards TR (10 μM) at different enrichment times.

**Figure 6 nanomaterials-13-00239-f006:**
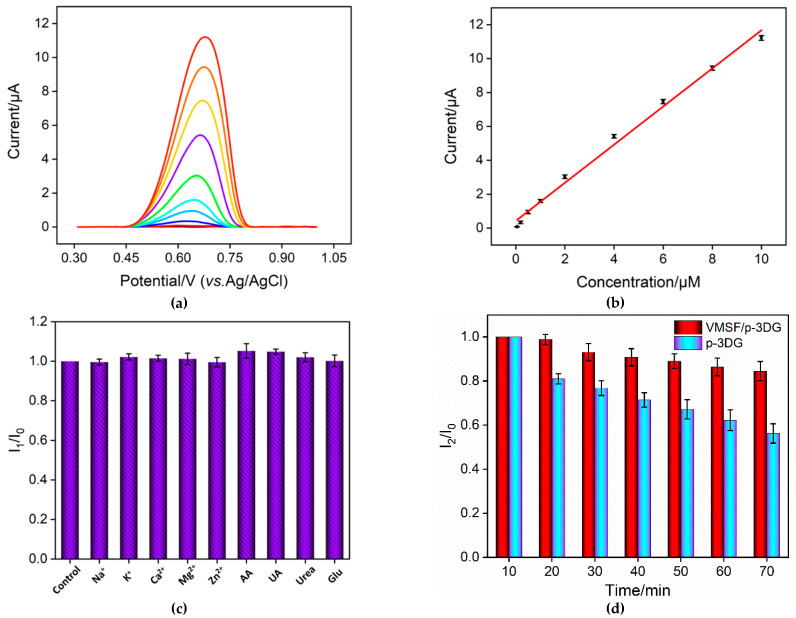
(**a**) DPV response of VMSF/p-3DG electrode with different concentrations of TR in PBS (0.01 M, pH 7.4). (**b**) The corresponding calibration curves. (**c**) The relative peak current ratio (I_1_/I_0_) in presence of different interfering species. I_0_ and I_1_ depict the peak currents measured in TR solution (2 μM) in the absence and presence of interfering species (100 μM). (**d**) The relative peak current ratio (I_2_/I_0_) obtained in successive scan in urine (diluted by a factor of 10) containing TR (10 μM). I_0_ and I_2_ depict the peak currents measured at the indicated time.

**Figure 7 nanomaterials-13-00239-f007:**
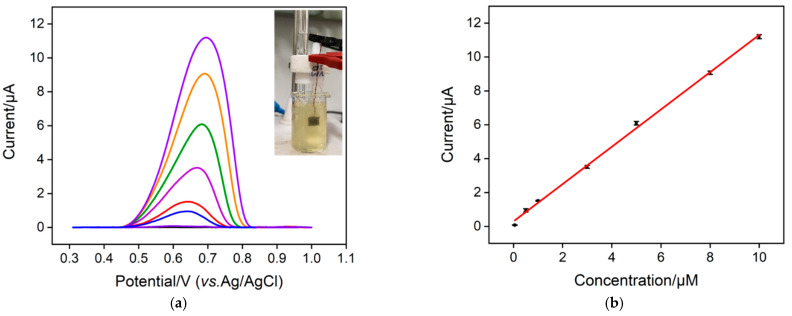
(**a**) DPV responses of VMSF/p-3DG electrode to various concentrations of TR in urine (diluted by a factor of 50). (**b**) The corresponding calibration curves. Error bars denote the relative standard deviation (RSD) of three measurements.

**Figure 8 nanomaterials-13-00239-f008:**
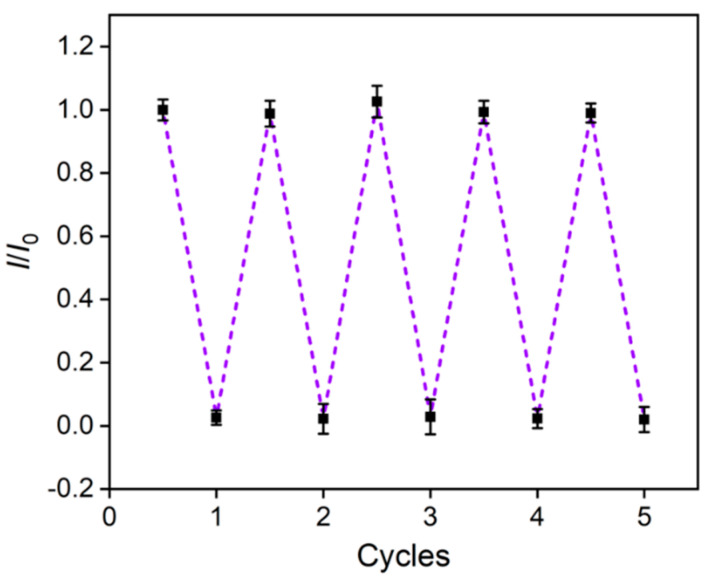
The relative peak current ratio obtained in the original or reproduced sensor. The above five points were obtained in TR solution and the bottom five points were produced in the supporting PBS electrolyte.

**Table 1 nanomaterials-13-00239-t001:** Comparison of the preformation for detection of TR using different electrochemical sensors.

Electrode	Method	Linear Range (μM)	LOD (μM)	Ref.
NGO/SPCE	DPV	0.04–151.6	0.004	[50]
P-MCO/GCE	DPV	0.5–1415.8	0.047	[51]
ZIF-67/Bio-MCM-41/CQDs/GCE	DPV	0.06–69.76	0.031	[52]
FeV NPs/SPCE	DPV	0.02–122.1	0.008	[53]
Bi/PSi/CNTPE	DPV	0.1–260	0.03	[54]
CILE	DPV	0.25–100	0.05	[55]
BDDE	DPV	0.2–40	0.12	[56]
VMSF/p-3DG	DPV	0.05–10	0.03	This work

NGO: spherical-like NiO@Gd_2_O_3_; P-MCO: MgCo_2_O_4_; ZIF-67: zeolitic imidazolate framework-67; Bio-MCM-41: bio-mobile crystalline material-41; CQDs: carbon quantum dots; GCE: glassy carbon electrode; FeV NPs: iron vanadate nanoparticles; SPCE: screen-printed carbon electrode; Bi/PSi: bismuth@porous silicon; CNTPE: carbon nanotubes paste electrode; CILE: carbon ionic liquid electrode. BDDE: boron-doped diamond electrode.

## Data Availability

The data presented in this study are available on request from the corresponding author.

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
