# Peer review of "Equipment of Vertically-Ordered Mesoporous Silica Film on Electrochemically Pretreated Three-Dimensional Graphene Electrodes for Sensitive Detection of Methidazine in Urine"

_nanomaterials, 2023, doi:10.3390/nano13020239_

Round 1

Author Response

Reviewer #1, General Comment:In this manuscript, the author developed a vertically-ordered mesoporous silica film (VMSF) on electrochemically preactivated three-dimensional (3D) graphene electrode (p-3DG), which can achieve direct and sensitive determination of methylthiopyridazine (TR) in urine. The developed materials seem interesting for electrochemical sensors. This manuscript can be accepted after a few major modifications.

Answer: We thank the reviewer for the positive and constructive comments. Accordingly, revision with more discussion and correction has been made. We hope that the reviewer would now find this revision acceptable for publication.

Reviewer #1, Comment 1:The author used lots of acronyms in the abstract, and some of them are used only once (ex:EASA), which is not needed.

Answer: We thank the reviewer for the helpful comment. As suggested, the acronyms in the abstract have been revised in the revision.

Reviewer #1, Comment 2:Please clarify the novelty issue by comparing it with more recent papers.

Answer: We thank the reviewer for the constructive comment. According to the reviewer’s suggestion, the novelty of the work has been added by comparing it with more recent papers. The following sentences have been added in section 3.1 of the revision: “Compared with the surface treatment of 3DG electrode with air plasma to enhance hydrophilicity, this method does not require special instruments or complex operations [48]. Compared with the strategy of using 3-aminopropyltriethoxysilane as the binding layer to achieve stable equipment of VMSF on the 3DG surface [49], this method does not require a non-conductive bonding layer. Therefore, the strategy of pre-activating 3DG using simple electrochemical polarization has the advantages of simple method, easy operation, improved hydrophilicity and electrocatalytic activity of 3DG, and direct growth of VMSF on p-3DG without adhesive layer.”

Reviewer #1, Comment 3:For the reader, the placement of the figures in the manuscript is complicated. Physical and electrochemical characterization should be in separate sections for better understanding.

Answer: We thank the reviewer for the constructive comment. Accordingly, the placement of the figures in the manuscript has been revised in the revision. The physical and electrochemical characterization have been placed in separate sections for better understanding.

Reviewer #1, Comment 4:In CV, the oxidation peak is located at 0.8 V, but in DPV, it is at around 0.68 V. Why is that?

Answer: We thank the reviewer for the comment. In the cyclic voltammetry (CV) measurement, the potential on the working electrode changes linearly at a constant scanning rate, and the corresponding current curve is recorded. The peak current measured by CV includes charging current and Faraday current. Unlike CV, differential pulse voltammetry (DPV) employs the superposition of linearly increased voltage and rectangular pulse with constant amplitude. Two currents before and after the pulse period are measure and the electrolytic current during the fast pulse period is obtained by subtracting the two currents. Thus, the peak potentials in CV and DPV curves are different owing to the different voltage modes. Usually, the peak potential in DPV curve corresponds to the potential with the maximum slope of current in CV curve. In Figure 5a and b of the revison, the peak potential in DPV curve (0.68 V) is also corresponds to the value (0.68 V) near the maximum slope of oxidation peak in CV curve.

Reviewer #1, Comment 5:Is the electrode material exhibiting a diffusion control process? Need experiment.

Answer: We thank the reviewer for the constructive comment. The diffusion control or adsorption control process of the electrode material can be easily investigated by measuring CV curves in standard redox solution with different scan rate. However, it is very regrettable that we will spend up to 50 days of winter vacation to control the rapid spread of Convid-19 in China. It's a pity that we can't finish the experiment at present. We will be highly appreciated if the reviewer can understand our difficulty.

Reviewer #1, Comment 6:The author didn’t show any stability test.

Answer: We thank the reviewer for the constructive comment. The stability of developed sensor in successive measurement in complex matrix is investigated and results are demonstrated in Figure 6d. As shown, 85% of the initial signal is still retained on VMSF/p-3DG electrode for the 7th measurement in urine that was only diluted by a factor of 10. In addition, the electrochemical response of the developed sensor towards TR (1 µM) is 97.3% (n=3) after 7-day storage in air at room temperature, indicating high stability in storage (See section 3.6 of the revision).

Reviewer #1, Comment 7:It is suggested that the author compare their sensor performance with recently published research in a comparison table and include the results in the revised manuscript.

Answer: We thank the reviewer for the helpful comment. Accordingly, Table 1 has been added in the revision to compare the sensor performance with recently published researches. In addition, the corresponding discussion has also been added (See the 1st paragraph of Section 3.5 of the revision).

Reviewer 2 Report

Authors have presented an article entitled “Equipment of Vertically-Ordered Mesoporous Silica Film on Electrochemically Pre-treated Three-Dimensional Graphene Electrodes for Sensitive Detection of Methidazine in Urine” Though the manuscript is well written and organized but there is scope for further improving the quality of the draft before considering for publication.  

Few minor comments are listed below:

1. As author mentioned that in Figure 2e, the deposition of graphene network, then how it is smooth in appearance? Please describe.

2. There are some lamps in Figure 2f as well cracks, what it means. Please explain briefly.

3. The y axis of Figure 7, it not clear. Is it I/I0. Check and rectify.

4. Authors nicely designed the results and discussion section, they should check the typos and grammatical mistakes in the revised version.  

Author Response

Reviewer #2, General Comment:Authors have presented an article entitled “Equipment of Vertically-Ordered Mesoporous Silica Film on Electrochemically Pre-treated Three-Dimensional Graphene Electrodes for Sensitive Detection of Methidazine in Urine”. Though the manuscript is well written and organized but there is scope for further improving the quality of the draft before considering for publication.

Answer: We thank the reviewer for the positive and constructive comments. Accordingly, revision with more discussion and correction has been made. We hope that the reviewer would now find this revision acceptable for publication.

Reviewer #2, Comment 1:As author mentioned that in Figure 2e, the deposition of graphene network, then how it is smooth in appearance? Please describe.

Answer: We thank the reviewer for the constructive comment. Three-dimensional graphene (3DG) employed in this work is monolith of graphene foam grown via chemical vapor deposition (CVD) using macroporous Ni foam as the template. After graphene growth, the nickel substrate could be removed by overnight incubation with hot HCl solution. Thus, 3DG exhibits a monolithic and well-defined macroporous structure. And the surface of graphene skeleton is smooth in a large scale. To show the wrinkled structure of graphene, high-resolution scanning electron microscope (HRSEM) image of 3DG has been added as inset of Figure 2e in the revision.

Reviewer #2, Comment 2:There are some lamps in Figure 2f as well cracks, what it means. Please explain briefly.

Answer: We thank the reviewer for the helpful comment. As shown in Figure 2f, the surface of p-3DG becomes rough with a few cracks. These cracks might be resulted from the anodic etching of graphene at high potential in electrochemical polarization. Owing to the 3D structure, different positions of 3DG have different heights. Since the SEM image is obtained by focusing to a certain height, the brightness displayed on the photo varies from place to place due to the different heights of different areas. On the other hand, the conductivity of the defect area might also change. Thus, some bright areas, such as lamps, will appear in the SEM image.

Reviewer #2, Comment 3:The y axis of Figure 7, it not clear. Is it I/I0. Check and rectify.

Answer: We thank the reviewer for the careful reading and helpful comment. Accordingly, the unclear y axis has been corrected in the revision.

Reviewer #2, Comment 4:Authors nicely designed the results and discussion section, they should check the typos and grammatical mistakes in the revised version.

Answer: We thank the reviewer for the careful reading and constructive comment. According to the reviewer’s suggestion, the typos and grammatical mistakes have been corrected in the revision.

Reviewer 3 Report

The present manuscript demonstrates the fabrication of a pre-treat 3DG (p-3DG) electrode followed by the growth of vertically-ordered mesoporous silica film (VMSF) by using CTAB and TEOS. The grown electrode due to its high surface area and resistance to negative ions demonstrated better performance for the electrochemical detection of methylthiopyridazine (TR) in urine. The authors have characterized the materials using various techniques such as contact angle, SEM, TEM, and XPS demonstrating the well-constructed hydrophobic surface and nanomorphology. The electrode displayed a very good electrocatalytic detection response with a nanomolar sensitivity. Finally, an actual urine sample with the presence of several proteins and ions also displayed a linear detection response. The finding presented in the manuscript are appealing and can be published after addressing minor concerns.

1.  In the section “Preparation of VMSF/p-3DG electrode” how authors have decided that “-350 μA was applied for 10 s” will be the best fabrication conditions.

2.  The growth of silica on graphene is increasing the area of the electrode, not the conductive surface area, then how the detection limit was increased. Do the authors have any data for the electrochemical surface area of the electrode? Also, if it is feasible, please provide the BET surface area.

3.  Can the authors comment on the growth mechanism of these perfect hexagonal silica structures? 

Author Response

Reviewer #3, General Comment:The present manuscript demonstrates the fabrication of a pre-treat 3DG (p-3DG) electrode followed by the growth of vertically-ordered mesoporous silica film (VMSF) by using CTAB and TEOS. The grown electrode due to its high surface area and resistance to negative ions demonstrated better performance for the electrochemical detection of methylthiopyridazine (TR) in urine. The authors have characterized the materials using various techniques such as contact angle, SEM, TEM, and XPS demonstrating the well-constructed hydrophobic surface and nanomorphology. The electrode displayed a very good electrocatalytic detection response with a nanomolar sensitivity. Finally, an actual urine sample with the presence of several proteins and ions also displayed a linear detection response. The finding presented in the manuscript are appealing and can be published after addressing minor concerns.

Answer: We thank the reviewer for the positive and constructive comments. Accordingly, revision with more discussion and correction has been made. We hope that the reviewer would now find this revision acceptable for publication.

Reviewer #3, Comment 1:In the section “Preparation of VMSF/p-3DG electrode” how authors have decided that “-350 μA was applied for 10 s” will be the best fabrication conditions.

Answer: We thank the reviewer for the constructive comment. According to the reviewer’s suggestion, the related reference for the fabrication process (Ref. 44) has been added in the revision. In addition, the deposition condition has been adjusted to current density rather than current value in the revision.

Reviewer #3, Comment 2:The growth of silica on graphene is increasing the area of the electrode, not the conductive surface area, then how the detection limit was increased. Do the authors have any data for the electrochemical surface area of the electrode? Also, if it is feasible, please provide the BET surface area.

Answer: We thank the reviewer for the careful reading and the helpful comment.

  • As pointed by the reviewer, the wrong description has been corrected.
  • The improvement of the detection limit is resulting from the electrostatic enrichment effect of VMSF nanochannels towards TR. Briefly, VMSF are rich in silanol group, that has pKa of about 2. Thus, the surface of VMSF nanochannel is negatively charged in conventional pH solutions, which can generate electrostatic attraction towards positively charged small molecules. As the pKa of TR is ~9.5, it is positively charged in the electrolyte. Thus, TR can be enriched by negatively charged nanochannels of VMSF, leading to high detection sensitivity. This enhanced electrochemical performance of VMSF/p-3DG towards TR is proven in Figure 5 a and b in the revision.
  • The change of electrochemical surface area can be observed in CV curves obtained on different electrodes in standard redox probe (Figure 1a, Figure 3 a and b). It is very regrettable that we can't measure the electrochemical surface area or BET data now because of the long winter vacation to control the rapid spread of Convid-19 in China.

Reviewer #3, Comment 3:Can the authors comment on the growth mechanism of these perfect hexagonal silica structures?

Answer: We thank the reviewer for the constructive comment. According to the reviewer's suggestion, the growth mechanism of VMSF has been added in the revision (See the 2nd paragraph of section 3.1 of the revision).

Round 2

Reviewer 1 Report

The author has addressed all the comments. It can be accepted in its present form.